# The effect of weight loss interventions in truck drivers: Systematic review

**Elizabeth K. Pritchard**[ID]◐*, **Hyunjin Christina Kim**◐, **Nicola Nguyen**◐, **Caryn van Vreden**‡, **Ting Xia**‡, **Ross Iles**[ID]◐

Department of Epidemiology and Preventive Medicine, Monash University, Melbourne, Australia

◐ These authors contributed equally to this work.
‡ These authors also contributed equally to this work.
* elizabeth.pritchard@monash.edu

## Abstract

## Introduction

Truck driving is the most common vocation among males internationally with a high proportion overweight/obese due to a combination of work and lifestyle factors leading to health complications. With limited studies in this area, this systematic review aimed to identify and describe interventions addressing weight reduction in truck drivers.

## Methods

Five electronic databases were searched, January 2000 to June 2020 (CINAHL, Cochrane Library, Embase, Ovid MEDLINE, Scopus). Inclusion criteria: experimental primary studies, long-distance (≥500 kms) truck drivers, peer reviewed publications in English. Weight loss interventions included physical activity, diet, behavioral therapy, or health promotion/education programs. Exclusions: non-interventional studies, medications or surgical interventions. Two independent researchers completed screening, risk of bias (RoB) and data extraction with discrepancies managed by a third. Study descriptors, intervention details and outcomes were extracted.

## Results

Seven studies (two RCTs, five non-RCTs,) from three countries were included. Six provided either counselling/coaching or motivational interviewing in combination with other components e.g. written resources, online training, provision of exercise equipment. Four studies demonstrated significant effects with a combined approach, however, three had small sample sizes (<29). The effect sizes for 5/7 studies were medium to large size (5/7 studies), indicating likely clinical significance. RoB assessment revealed some concerns (RCTs), and for non-RCTs; one moderate, two serious and two with critical concerns. Based on the small number of RCTs and the biases they contain, the overall level of evidence in this topic is weak.

**Data Availability Statement:** Data attached in Supplementary file.

**Funding:** RI, EP, CvV, TX. National Health and Medical Research Council https://www.nhmrc.gov.au/ grant number GNT1169395 (RI) The Transport

Workers Union https://www.twu.com.au/ Linfox
https://www.linfox.com/- Linfox provided partial
salary support for Ross Iles, Elizabeth Pritchard,
Caryn van Vreden and Ting Xia Centre for Work
Health and Safety https://www.centreforwhs.nsw.
gov.au/ The funders had no role in study design,
data collection and analysis, decision to publish, or
preparation of the manuscript.

**Competing interests:** Even though Linfox is a
commercial funder, this does not alter our
adherence to PLOS ONE policies on sharing data
and materials.

## Conclusion

Interventions that include a combination of coaching and other resources may provide successful weight reduction for truck drivers and holds clinical significance in guiding the development of future interventions in this industry. However, additional trials across varied contexts with larger sample populations are needed.

## Introduction

Truck driving is the most common occupation among males in 29/50 of the United States [1] and also across Australia [2] with an estimated number of drivers as 3.2 million in Europe (2015) [3], approximately 2.8 million working in the U.S. in 2018 [4], and approximately 200,000 in Australia [5, 6]. The nature of this occupation increases the risk of chronic diseases due to work, environmental, social and personal factors [7]. In the US >85% of truck drivers are male with a median age of 46 compared with national average of workers at 41 years adding additional health risks and issues around attrition of workforce as they age [8]. In Australia, the average age of truck drivers is 44.3 compared to the average worker at 39 years [9] with 6.5% reported as female [10]. This aging and predominately male dominated population are exposed to long working hours, shift-work, sleep deprivation, sedentary roles, social isolation, and limited access to healthy foods and exercise facilities [11–13]. As a result, these elements can have negative health consequences for drivers, and can explain in part the high incidence of obesity amongst truck drivers [12, 13]. In a systematic review looking at health and wellbeing of drivers, over nine studies found that more than 50% of their sample populations were obese [7] with one reporting 83.4% of the 316 truck drivers being obese [14]. A cross-sectional survey of 231 Australian truck drivers found that almost 90% of drivers were overweight or obese [15]. This is a rate of obesity that is nearly 1.4 times that of the general male population across similar age groups [16].

Health complications including cardiovascular disease, diabetes, obstructive sleep apnea (OSA), osteoarthritis, and cancers are positively associated with being overweight [17]. These in turn can have a negative impact on driving performance and safety [12, 13]. This includes crashes, where drivers with obesity were reported as being twice as likely to crash than drivers in the normal weight range [18]. Furthermore, drivers who were overweight were more likely to have OSA and therefore more likely to fall asleep behind the wheel, increasing their crash risk, injury risk and overall level of health [19].

Food options at most truck stops across Australia have been identified as 'unhealthy' with high fat and carbohydrate content [15]. The cross-sectional survey found that 63% of the truck drivers consumed at least one serving of 'unhealthy' food each day [15]. Approximately 80% of these participants also failed to meet the National guidelines and recommendations for physical activity per week [15]. The health risks of sedentary occupations are well documented and include obesity, cardiovascular disease, cardiorespiratory problems, diabetes and even cancer which creates cyclical health risks [20]. To combat this growing trend, the first line management of weight and maintenance of health needs to include modifications to behavior to consistently choose a healthy diet and increase level of physical activity [21]. The combination of reducing caloric intake through healthy food choices (low in sugars and fats) and increasing energy expenditure through physical activity, can tip the balance favoring a calorie deficit and subsequently lead to weight loss. It is important to identify lifestyle factors, such as diet and physical activity, that may contribute to the management of truck driver weight in order to

identify interventions to combat the negative impact. Behavior modification, such as self-monitoring and effective goal implementation, can further help a person to reach and maintain their healthy body weight [21, 22].

A meta-analysis of 45 studies investigating weight loss interventions focusing on both food intake and physical activity with adults who were obese [23] showed that participants with effective and sustained behavioral strategies were less likely to regain their lost weight. However, they also showed no evidence of effectiveness when focusing on diet or physical activity alone. These modifications can be created through a combination of education, social support, and counselling [21]. Motivational interviewing (MI) is an example of a person-centered counselling strategy where a health coach works with the individual to identify their readiness for change and supports them through achieving the transition process "by exploring and resolving client ambivalence" [24, 25]. The coach helps the participants set realistic goals and build achievable steps towards each goal through reflective listening, asking open-ended questions, ascertaining the person's readiness to change and embracing the use of 'change talk' [26]. MI coaching has been shown to enhance weight loss in individuals who were obese across multiple systematic reviews [24, 27, 28] however, a review of interventions to address weight reduction with truck drivers has not yet been completed.

Therefore, the primary objective of this systematic review was to explore interventions for weight reduction in truck drivers and identify which interventions were effective for weight reduction in truck drivers to inform a future pilot program in Australia.

## Methods

Five electronic databases were searched including CINAHL, Cochrane, Embase, Ovid Medline, and SCOPUS with the following MeSH search terms used e.g. 'truck driver' or 'automobile driver' or 'motor vehicles' (S1 Appendix). The reference lists of 'key' systematic reviews were manually searched for any additional studies that met the inclusion criteria. This review protocol was registered in Prospero CRD42020213926.

Inclusion criteria were peer reviewed experimental and quasi-experimental primary studies involving long-distance (≥500 kms per day) truck drivers who were ≥18 years. Studies published in English (as translation resources were unavailable) in journals from January 2000 to June 2020 were included to ensure the most up to date interventions were captured. Where studies included other transport drivers (e.g. train, bus), a minimum of 50% of study participants had to drive trucks. Weight loss interventions included physical activity, diet therapy, behavioral therapy, and health promotion or education programs. Outcomes had to directly measure body weight or some measure related to body composition related to weight loss (e.g. body measurements such as body mass index (BMI), waist-to-hip ratio (WHR), skin-fold thickness), or impact on long-term health conditions associated with obesity e.g. Type-2 diabetes, or Metabolic Syndrome (MeS), that raises the risk of heart disease. Exclusion criteria were non-interventional studies, grey literature and weight loss strategies that used medications or surgical intervention such as gastric bypass or banding and sleeve gastrectomy.

Title and abstracts were screened independently by two reviewers (CK, RI) then full text reviews were completed (CK, EP). If consensus was not reached, a third reviewer was approached (RI) (S2 Appendix). Studies reporting the same cohort were merged as per the Cochrane protocols for systematic reviews [29].

Data were extracted independently onto a Microsoft Excel spread sheet (refer Supporting Information 1: Data extraction Excel spreadsheet) by two reviewers (CK, EP) to identify the interventions, population groups, and outcomes of each study. All outcome measurements (weight, BMI, fat mass, body measurements), were converted to kilograms (kg) and

centimeters (cm) for ease of comparison, reporting effect sizes between baseline and reassessment and/or follow-up if provided.

### Risk of bias assessment

The studies were assessed using Cochrane's assessment tools; Risk of Bias 2 (RoB 2) for randomized controlled trials (RCTs) [30] and Risk of Bias in Non-randomized Studies—of Interventions (ROBINS-I) tool for non-RCTs [31].

All assessments were completed independently by two people (CK, EP or CK, NN). RoB 2 has five criteria of bias: 1. Arising from the randomization process; 2. Due to deviations from intended interventions; 3. Due to missing outcome data; 4. In measurement of the outcome; and 5. In selection of the reported result. Each bias criterion is rated as either low risk, some concerns or high risk when using the tool logarithms and determined by these criteria. The Microsoft Excel RoB 2 tool from Cochrane was used for each study, as there were only two outcomes that were common across two studies [32]. All eligible studies were discussed in this review regardless of their RoB results as there is limited work in this area. Sub-group analysis is presented where possible.

The ROBINS-I includes seven domains of bias: 1. Due to confounding; 2. In selection of participants into the study; 3. In classification of interventions; 4. Due to deviations from intended interventions; 5. Due to missing data; 6. In measurement of outcomes; and 7. In selection of the reported result. Ratings criterion included no information provided, low, moderate, serious, or critical risk of bias as determined by these criteria.

Each paper was scored using the appropriate tool by two independent researchers (CK, NN) with a third for consensus if required (EP).

### Synthesis of data

Meta-analysis of the data was not able to be performed due to the different study designs and level of outcome reporting. The Synthesis Without Meta-analysis (SWiM) framework was used to guide the synthesis of data [33]. Groupings were study design (RCTs and quasi RCT; pre-post studies); outcomes including weight, BMI, fat mass, and measurements; the method and length of intervention delivery; and the target group. Where possible, the standardized metric effect size (Cohen's d) of the intervention was reported from the study or calculated where possible, to enable comparison of intervention effects across the different outcome measures applied [34]. The d statistic was interpreted as 0.2 representing a small effect, 0.5 a medium effect and 0.8 and higher a large effect [34]. For studies where effect size was unable to be calculated $p$ values, median and interquartile range were reported. Criteria used to prioritize the findings were study design (those with a control) and those where risk of bias assessment was either low, moderate or some concerns. Those with high or critical risk of bias concerns is not discussed in detail. Investigations for heterogeneity were not prespecified prior to analysis as the breadth of data was not yet determined in this area. Findings have been presented in tables (key characteristics and outcomes) and figures (risk of bias findings). A description of the synthesis of findings is presented and related to answering the research question. Limitations of the study and synthesis is also reported.

## Results

### Literature search

A total of 422 articles were obtained from the database search (Fig 1) and two from the reference list search, with 407 remaining after 17 duplicates were removed. Following abstract

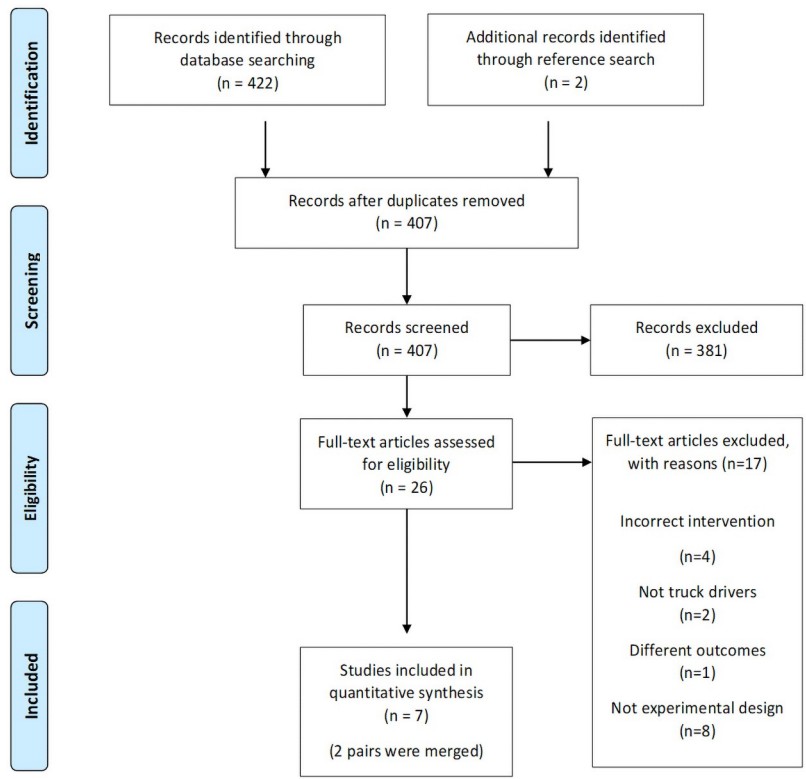

**Fig 1. PRISMA flow of studies included in review.**

screening and full text review, a total of nine articles met the inclusion criteria. Two studies were of the same cohort with a 30-month follow-up and were merged [26, 35], and two were the same cohort with a different slant on reporting the outcomes so were merged [36, 37]. This left seven studies for analysis [6, 26, 37–41]. One was an RCT [37], one was a cluster RCT [6], one was a quasi-experimental intervention [38] while the remaining four were single group pre- and post-test design [26, 39–41]. Five were conducted in the U.S. [6, 26, 38–40], one in Australia [41], and one in Finland [37].

## Level of evidence

The risk of bias appraisal revealed that there were some concerns, moderate or critical levels of bias in all studies. For the two RCTs (Fig 2), one did not fully identify the randomization process, [6], the other did not state if the analysis conducted was congruent with the pre-planned analysis, and did not report one of the areas they assessed (sleep) [37], and both did not state the specifics around blinding. For the non-RCT studies (Fig 3), two were assessed as having a critical level of bias, with the first due to missing data from the self-assessment tool [26], and the second due to high attrition bias, missing data from self-assessment and intervention deviation [41]. Two were assessed as serious with missing data from the self-reported measures [38, 39], and one lacked specificity around randomization, outcomes and results [40].

## Details of outcomes and interventions

A total of 1214 participants were included across all studies at baseline including four studies with ≤46 participants, one with 113 and two >400. There were moderate to high levels of

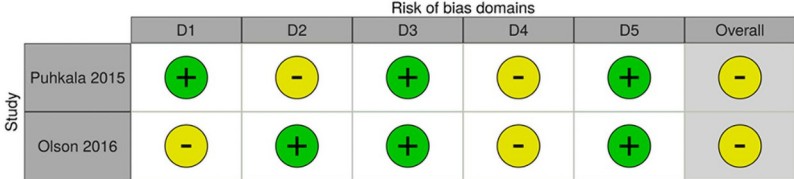

Fig 2. Risk of bias assessment for the RCTs completed. Domains: D1: Bias arising from the randomization process; D2: Bias due to deviations from intended intervention; D3: Bias due to missing outcome data; D4: Bias in measurement of the outcome; D5: Bias in selection of the reported result.

attrition (23–49%) at follow-up (Table 1). The length of intervention varied from one month [40] to 1- months [37] with follow up times of 6-months [38], 24-months [37] and 30-months [35]. Outcomes measured varied across studies from weight, BMI, waist and fat mass measurements and reported as described in the included study's findings (Table 2). Weight was the most commonly measured outcome in 6/7 studies [6, 26, 37–40], BMI in 4/7 [6, 26, 39, 40], waist measurement 3/7 [6, 37, 39], and fat mass 1/7 [37]. Two RCTs [6, 26] and one pre-post study [40] showed a large effect on the specific measured outcomes, with the others showing a small to medium effect only [37, 39] and one providing insufficient information to calculate [38]. These results were similar for BMI. For waist and fat measurement a large effect was found in one study only at 12-months, but not 24-months [37]. The majority of the effects observed are medium to large size, indicating likely clinical significance.

Four studies explored additional health risk factors relating to weight i.e. two measured blood glucose levels with readings identified as 95.81 (mg/dl) pre and 110.44 post, both of which were in the normal range (not significant at $p = 0.46$) with no follow-up testing done at 30 months [26], and the other reported 123 (mg/dl) at baseline decreasing to 98 on exit (not

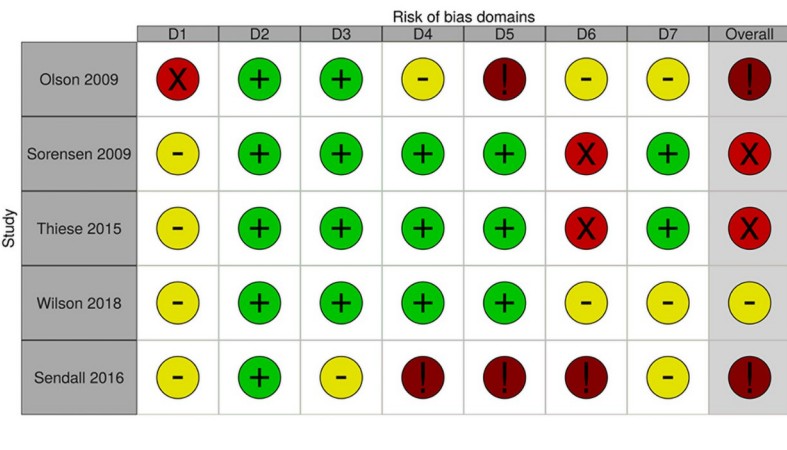

Fig 3. Risk of bias assessment for the non-RCTs completed. Domains: D1: Bias arising from the randomization process; D2: Bias due to deviations from intended intervention; D3: Bias due to missing outcome data; D4: Bias in measurement of the outcome; D5: Bias in selection of the reported result.

**Table 1. Characteristics of studies included in the review.**

| Study | Country | Setting (sample inclusion criteria), n = | Intervention type | Method of intervention delivery | Intervention Timeframe | Length and frequency | Measurement timeframes |
|---|---|---|---|---|---|---|---|
| **Olson et al. (2009) Single group pre- and post-test design Pilot study** *Merged with*: **Wipfli et al. (2013) Follow-up study** | USA | Trucking carriers in Pacific Northwest region of the USA n = 29 Follow up n = 15 | SHIFT program (Safety & Health Involvement for Truckers) • Motivational interviewing (MI) • behavioral computer-based training with assignments and pre-post- tests • weight loss/safety team competitions • behavioral self-monitoring • Follow-up interview | Individual mainly, Online, One-on-one phone (must be parked up), Team competition | 6-months | 4 MI sessions (30–45 mins each) 4 units computer-based training | Baseline (T1) and end of program (T2) 30 months follow-up (t3) |
| **Olson et al. (2016) Cluster RCT** | USA | Interstate truck drivers BMI ≥27, interest in losing weight No medical conditions prohibiting increased physical activity Intervention n = 229 Control n = 223 Total n = 451 | SHIFT program, updated • behavioral computer-based training with assignments and pre-post- tests • weight loss/safety team competitions • behavioral self-monitoring • MI from 4 female coaches | Individual mainly, Online, One-on-one phone (must be parked up), Team competition | 6-months | 1–4 calls Time not stated | Baseline and end of program No follow-up |
| **Puhkala et al. (2015)** *merged with* **Pukhala et al. (2016) RCT (intervention 1 full 12 months (LIFE) and Intervention 2 delayed 3 months (REF))** | Finland | 30–62 yo male truck or bus drivers Waist circumference ≥100cm Intervention LIFE n = 55 Intervention REF n = 58 Follow-up 12 months LIFE n = 47 REF n = 48 Follow-up 24 months LIFE n = 37 REF n = 43 | Lifestyle counselling/ goal setting approach on nutrition, physical activity and sleep | Individual counselling/ education sessions, one-on-one phone and face-to-face (place not stated) | 12-months (LIFE group) 3-months (REF group) | LIFE group—6x 60 mins counselling & 7x 30 mins with nutritionist or PT (12-month intervention) REF group 2x face to face counselling (time not stated) & 3 phone contact sessions (3-month intervention —began after the LIFE group had finished) | Baseline and 12-months 24-month follow-up |
| **Sendall et al. 2016 Single group pre- and post-test design** | Australia | Baseline n = 46 End of program n = 22 | 7 options 3–4 selected by each workplace: Posters; healthy vending machines; free fruit; 10,000 steps challenge; health eating or physical activity talks; health messages; Facebook webpage | Health promotion population-based approach in workplace, social media messages (for drivers) | 6-months | Not stated | Baseline and end of program No follow-up |

*(Continued)*

**Table 1.** (Continued)

| Study | Country | Setting (sample inclusion criteria), n = | Intervention type | Method of intervention delivery | Intervention Timeframe | Length and frequency | Measurement timeframes |
|---|---|---|---|---|---|---|---|
| **Sorensen et al. (2009)** **Non-randomized, control group design** **Quasi-experimental** | USA | Unionized truck drivers and dock workers Intervention group n = 227 Control n = 315 Total n = 542 Follow-up n = 405 total | • MI (Telephone-delivered health promotion on smoking cessation and weight management) • Personalized health messages on diet, exercise, lifestyle habits • Additional resources mailed out | One-on-one counselling via phone, resources | 4-months | 1–5 phone calls duration not stated | Baseline and 10-months 6-months follow-up |
| **Thiese et al. (2015)** **Single group pre- and post-test design** | USA | Long-haul commercial motor vehicle driver BMI $\geq$30kg/m$^2$ $\geq$21 yo n = 13 | • Health education materials on healthy diet, exercise • Exercise equipment • Portable stove, pans and cook book • Telephone based health coaching | Individual, Phone based, Equipment given | 3 months | 12 calls (weekly) with the health coach (time & length made by the driver) | Baseline and weekly testing after week 2 (11 weeks) No follow-up |
| **Wilson et al. (2018)** **Single group pre- and post-test design** | USA | $\geq$18 yo drivers at a global battery manufacturing plant n = 19 | • MI from the project implementer (PI) • Education and materials | Individual, Face-to-face and phone at the health clinic | 1 month | 2 x meeting & 1x Phone call with PI 2 hours overall for the month | Baseline and end of program No follow-up |

*Effect size: 0.2 small effect; 0.5 medium effect; 0.8 and higher large effect.

significant at p = 0.79) [39]. One reported the presence of diabetes (type not stated) with control at 13% at baseline, the intervention group at 12% and blood glucose risk with slight increase across both groups at 6 months which was not significant (p = 0.84) [6], and one study reported the presence of Type-2 diabetes as 1% at baseline (no follow-up data) with over two-thirds (71%) having MeS at baseline which decreased in the two intervention groups at 12 months to 62% (in the LIFE group) and to 60% (in the REF group), also not significant (p = 0.34) [37].

Of the three studies with a follow-up period across 30-months, 24-months and 6-months respectively [26, 37, 38], only one intervention showed maintained weight reduction [26]. The 30-month follow-up paper demonstrated that the participants who completed the Safety and Health Involvement For Truckers (SHIFT) intervention continued to practice the habits learned and maintained their reduced body weight [35]. The second RCT identified a reduction in body weight for both groups assessed in the study (the Lifestyle counselling (LIFE) group receiving 12-months of counselling and the wait-list reference (REF) group receiving 3-months counselling). However, at follow-up occurring at 12 and 24-months, changes were not statistically significant [37].

Overall there was potential for medium to large effects on weight loss across all studies. This was demonstrated in the larger methodological studies (SHIFT) but required high levels of coaching input. Some of the smaller studies showed a large effect, but there is no indication of whether the size of the effect would be maintained in a larger trial. The SHIFT program had four evidence-based components:

**Table 2. Outcomes as reported in each study.**

| Study | Outcomes | | | | | | | |
|---|---|---|---|---|---|---|---|---|
| | Weight (kg) | Effect size[b] (d) | BMI | Effect size (d) | Waist measurement | Effect size (d) | Fat mass | Effect size (d) |
| **Olson et al. (2009)** *Merged with*: **Wipfli et al. (2013)** **Follow-up study** | [a] t1-t2: -3.5 (5.3 SD) **(p<0.01)** [a] t1-t3: -8.2 (7.2 SD) **(p<0.001)** | 0.68 med* 1.15 large* | [a] t1-t2−0.96 (1.5 SD) **(p<0.01)** [a] t1-t3: -2.7 (2.5 SD) **(p<0.001)** | 0.64 med* 1.16 large* | | | | |
| **Olson et al. (2016)** **RCT** | -3.31 **(p<0.001)** -7.29 Mean group diff [-9.76, -4.81 95% CI] | 1.22 large* | -1.00 kg/m² [-1.39, -0.62 95% CI] **(p<0.001)** | 1.25 large* | -0.76 Mean group diff [-1.25, -0.27 95% CI] | | | |
| **Puhkala et al. (2015)** **RCT** *merged with* **Pukhala et al. (2016)** | *12 months* LIFE -3.4 (6.6 SD) REF 0.7 (3.9 SD) (-6.2 to -1.9 95%CI) *24 months* LIFE -3.1 (9.0 SD) REF -2.5 (5.9 SD) [-3.8 to 2.9 95%CI] | 0.51 med* 0.18 small* 0.34 small | | | *12 months* LIFE -4.7 (5.8) REF -0.1 (3.6) *24 months* LIFE -4.5 (7.5) REF -4.4 (5.5) | 0.81 large 0.6 med | *12 months* LIFE -2.6 (5.1) REF 0.6 (3.4) [-4.9 to -1.4] *24 months* LIFE -2.2 (6.9) REF -2.3 (4.6) [-2.4 to 2.8] | 0.51 large* 0.18 small* 0.32 small |
| **Sendall et al. 2016** | | | Self-report 'obese' reduced 16% No other reported findings BMI or weight | unable to calculate | | | | |
| **Sorensen et al. (2009)** | *At follow-up* Int -0.03 Con +0.22 [0.62, 1.40 95%CI] (p = 0.74) | unable to calculate | | | | | | |
| **Thiese et al. (2015)** | *Post intervention 12 weeks* -3.2 **(p = 0.03)** | 0.16 small* | -1. kg/m² **(p = 0.03)** | 0.07 small* | 3.7 cm reduction 129.5 median (23.8 IQR) **(p = 0.06)** | Unable to calculate | | |
| **Wilson et al. (2018)** | -2.1 (1.4 SD) **(p<0.0001)** | 1.53 large* | -0.65 (0.44 SD) kg/m2 **(p<0.0001)** | 1.47 large* | | | | |

[a] t1 = pre-intervention; t2 = post-intervention; t3 = 30 month follow up.

[b] Effect size: 0.2 small effect; 0.5 medium effect; 0.8 and large effect.

* Statistically significant effect (p<0.05)

1. Behavioral computer-based training: Four units of content consisting of 20–45 min of interactive presentations including an overview of the SHIFT program, physical activity, diet and safety tutorials.

2. Weight loss and safe driving competition with incentives: Drivers were divided into teams competing to achieve the highest percentage of collective weight loss and fewest safety breaches. The winning team received financial incentives ranging from $10 to $1000.

3. Behavioral self-monitoring: Weekly logs of body-weight and number of days behavioral goals were met (e.g. diet control, physical activity, sleep). Drivers also received incentives for completing the first log, training and coaching calls, and again at the completion of the program.

4. MI: Trained coaches provided coaching sessions to the drivers (one-on-one). Sessions were held around weight loss behavioral goals, a change plan, and a summary with follow-up.

Drivers also received a step counter, technical support, and a resource book. Data were collected through intervention terminals (based in the company facilities) where they had one laptop in the drivers' lounge and another which could be borrowed for use on the road by participants.

The method and delivery of interventions ranged across all studies from use of printed resources, Facebook pages, vending machines with healthy food options, access to free fruit, competitions, online training, counselling, education, to MI coaching and counselling. The most commonly applied interventions were education or a form of counselling/coaching with six studies combining both [6, 26, 37–40]. Dietary education ranged from classes on healthy eating [41] computer-based training on diet, physical activity and risk management [6, 26], to providing drivers with a cookbook, portable stove and pans to cook on the road [39]. Physical activity education included a range of workout tutorials, through to providing the drivers with exercise equipment including yoga mats, pedometers, and dumbbells (Table 1). Six studies delivered a component of coaching or counselling [6, 26, 37–40]. Four of these utilized MI delivered by a range of trained clinicians [6, 26, 38, 40], the remaining two studies did not state the method of counselling/coaching [37, 39]. One study stated they enlisted four female coaches [6], however, the others did not specify who delivered the coaching.

Interventions were all targeted at the individual driver with 6/7 providing one-on-one sessions via the phone [6, 26, 37–40]; with two providing face-to-face sessions once at a health clinic [40] and one stated the counsellors traveled to the drivers [36]; two provided self-paced online training [6, 26]; one provided cooking and exercise equipment [39]; four provided health messaging and resources [38–41]; and two used the SHIFT program which included a team competition for weight loss and driver safety [6, 26]. Sendall et al. (2016) tested health promotion interventions (no coaching), to identify how these impacted the knowledge and behavior of drivers. Weight loss results varied across participants with only an overall reduction in self-reported BMI from the 'obese participants' identified [41].

The duration for coaching sessions ranged from an intended 30- to 60-minutes with the frequency of weekly to monthly (Table 1). Two of the seven studies were published by the same group of investigators using the same intervention [6, 26]. This group created and piloted an industry specific intervention, the SHIFT program [26], with the merged 30-month follow-up [35] and then progressed onto a RCT [6].

## Discussion

This systematic review has identified seven intervention studies conducted over the past 20 years that explored the effect of weight loss interventions on truck drivers. Only two were RCTs [6, 37] and just one of these showed significant results on reducing weight and BMI [6]. None of the four studies that measured incidence of diabetes or MeS, showed a significant reduction [6, 26, 37, 39]. Both used a multicomponent program incorporating a coaching/counselling approach of goal setting with individuals, and provision of information around healthy diets and physical activity. The difference between the programs were the styles of coaching. The one that showed a significant reduction in weight, delivered the SHIFT program which uses MI, computer-based training modules, and weight loss/safety competitions [6]. This appeared more successful than the counselling/coaching and information alone [37]. The quasi-experimental multi-component intervention study of MI, health messages and mailed out resources was not effective [38].

Two of the pre- and post-test studies also used MI combined with other educational components yielded significant results however the sample sizes were small with n = 29 [26] and n = 19 [40] and therefore the results need to be interpreted with caution. One other pre-and post-test study used a telephone coaching approach along with physical activity information and issued drivers with exercise equipment, with significant effects however the sample size (n = 13) was also very small [39]. The final study did not use any form of coaching [41]. The multicomponent studies provide an indication of what interventions may be effective depending, but this is likely to depend on the method of coaching and the components included.

Coaching is a common and effective approach used to elicit health behavior change [42, 43]. Effectiveness can depend on the framework used and also the skills of the clinician providing the sessions [44]. Coaching may range from telephone "check-in" calls with people to follow-up their progress [44], to health coaching or MI where the coach encourages the participant to reflect on their own barriers and provide solutions for progressing towards their goals [45]. The effectiveness of coaching interventions can also depend on the frequency and duration of the calls although there is no gold standard for this yet and is often underreported in primary studies [42, 44, 46]. Not all studies in this review describe these coaching interventions in detail and therefore it is difficult to compare the specifics of the program deliverables.

MI is considered an effective coaching approach for health behavior change [43, 47] and was the predominant approach described in 4/6 of the studies that used counselling [6, 26, 38, 40]. It is interesting to note that there were additional theoretical behavior change approaches described in weight loss literature which include (but not limited to) Self-determination Theory [48] (effective in maintaining weight loss) [49]; Social Cognitive Therapy, Transtheoretical Model, and Theory of Planned Behavior, all of which have been used to manage and maintain weight loss in adults [50]. Additional to the four studies that used a MI approach the other two used 'counselling' without stating their behavior change approach. As the MI approach was the most common across the included studies, MI is explored in more detail below.

A previous systematic review exploring changes in physical activity following MI, reported a small positive effect [51]. A more recent systematic review looked at the effects of health coaching and behavior modification with people with cardiovascular risk factors and identified a small but significant improvement on physical activity, dietary behaviors, health responsibility and stress management [43]. A large number of truck drivers have been diagnosed with cardiovascular disease and/or other chronic health conditions [11] and therefore a MI approach may be useful within the industry. Another review explored the effect of MI in adults through telehealth delivery and identified a greater weight loss experienced on 6 of 11 occasions compared to the no-treatment arm [27]. As truck drivers are on the road for many hours each day, the study supports the possibility of further trials using MI coaching with telehealth delivery, to positively impact weight and health promoting behaviors. Careful consideration of the challenges and potential inequalities in providing interventions for this group, such as possibilities of telehealth interventions, making sure the technology for any intervention is accessible, accessed safely and will work when they are travelling and remote, is required.

Sustainability of the positive effects of the interventions is also difficult to ascertain from the reviewed studies along with understanding the best method of recruitment for this population that is frequently on the road. Only three completed a follow-up study to assess sustained change [37, 38], with just one reporting positive results in weight and BMI reduction over time, using the SHIFT program [26]. However, attrition rates were relatively high. Ongoing studies investigating effective interventions for weight reduction with truck drivers may benefit from utilizing an MI approach in combination with online learning, competitions and resources as provided in the SHIFT program. Although gender sensitivities were not explored in the reviewed studies, this is another area to investigate in future research. Parallels between

other male dominated occupations and trucking, could be further explored to identify effective ways of encouraging men to participate in weight loss programs. The interventions described in this review may provide the way forward to improve health for truck drivers (both male and female) and potentially lead to long-term sustained improvements in weight reduction and therefore better health, however, there are still many unknowns about sustainability of changes in larger sample groups of people.

The risk of bias assessed from the reviewed studies ranged from some concerns to critical, which also impacted the findings of this review. The RCTs both had some concerns in the RoB assessment and the quasi-experimental and pre- and post-test studies had moderate, serious or critical concerns. The level of evidence and number of risks identified along with the limited sample sizes in all but two studies, needs to be considered. While each study contributes to our understanding of what may decrease weight and improve the health of truck drivers, the results of this review are inconclusive. Only one study was conducted in Australia and therefore it is also difficult to ascertain if these findings are generalizable to the Australian context.

A combined intervention as displayed in the SHIFT program [26] is the best evidence we could find from this review and may provide a way forward. Nevertheless, the feasibility and sustainability in the Australian transport context requires additional future trials.

The synthesis of findings from this review has identified there is a low level of evidence available to guide future interventions for effective weight reduction programs for truck drivers. These studies highlight the research gaps that still exist in the area of effective weight reduction programs for truck drivers. Well conducted clinical trials in larger sample groups are required to produce high-level evidence of effective interventions to reduce driver weight, applicability of interventions to the gender sensitivities of a predominantly male driver population, and consistency of measurement of outcomes to enable comparison across studies in the future. Ongoing research needs to focus on addressing these gaps to ameliorate the ongoing negative health and wellbeing implications of obesity for truck drivers.

The clinical significance of these results is important to note, as even a 1-kilogram reduction in weight can reduce the risk of diabetes by 16% [52]. With any reduction in risk of long-term health conditions developing, we are supporting the health and wellbeing of truck drivers. Only 2 studies reported a significant difference with a small effect (so there is the possibility that the effect is not clinically significant), whereas 5 of the differences identified were a medium to large effect, suggesting the differences observed are likely to be clinically significant and meaningful. Using a multi-pronged method of delivery as described in the studies with the most efficacious findings in this review, is more likely to yield effective long-term results, however, many of the studies were small in number with varied effect sizes and additional research is still required.

## Strengths and limitations

The strengths of this review were the specificity of the inclusion criteria, the number of databases explored, rigor in which the screening, RoB assessment and data extraction were conducted (reducing the risk of selection bias for this review) and the use of the structured approach to the analysis and synthesis without meta-analysis (SWiM). The SWiM framework allowed for a clearer description of the methods, provided clarity of the links of the synthesis and a checklist for how to group and report the findings. There were also no conflicts of interest for the authors.

One of the limitations is the narrow scope of this review exploring interventions to effect weight loss and not considering other chronic conditions e.g. Diabetes, Cardiovascular disease. Although these outcomes are very much interconnected in health, the scope of the review was

determined so as to inform a potential intervention pilot focusing on truck driver weight loss. We discovered there were few high-quality studies in this area, so caution must be applied in interpreting the findings. Along with the small sample sizes in many of the studies and high levels of attrition, this suggests a complexity of implementation factors that need to be carefully considered in future trials.

## Conclusion

A combination of MI and supporting resources has potential for long-term effectiveness in reducing truck driver body weight. However, the level of evidence in this area is minimal with only two RCTs available. Findings presented are also inconclusive due to the level of bias, small sample sizes, and designs of each of the studies included in this review. Further clinical trials are required with larger cohorts of truck drivers, and need to aim to include 12-month or longer follow-up periods, provide clear and detailed description of the intervention so they can be replicated elsewhere, use consistent measurement of weight reduction outcomes, and administer evidence-based interventions appropriate for the gender sensitivities within the industry. Future studies could then determine what interventions can be transferred to the Australian transport industry for weight reduction and how they could be sustainably implemented.

## Supporting information

**S1 Appendix. Ovid Medline search.**
(TIF)

**S2 Appendix. Data extraction excel spreadsheet.**
(XLSX)

## Author Contributions

**Conceptualization:** Elizabeth K. Pritchard, Caryn van Vreden, Ting Xia, Ross Iles.

**Data curation:** Elizabeth K. Pritchard, Hyunjin Christina Kim, Ross Iles.

**Formal analysis:** Elizabeth K. Pritchard, Ross Iles.

**Funding acquisition:** Ross Iles.

**Investigation:** Elizabeth K. Pritchard, Hyunjin Christina Kim, Nicola Nguyen, Ross Iles.

**Methodology:** Elizabeth K. Pritchard, Hyunjin Christina Kim, Ross Iles.

**Supervision:** Ross Iles.

**Writing – original draft:** Elizabeth K. Pritchard.

**Writing – review & editing:** Elizabeth K. Pritchard, Hyunjin Christina Kim, Nicola Nguyen, Caryn van Vreden, Ting Xia, Ross Iles.

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
