## [Decision Letter · Decision Letter 0]

13 Apr 2021

PONE-D-21-02019

The effect of weight loss interventions in truck drivers with obesity: Systematic review

PLOS ONE

Dear Dr. Pritchard,

Thank you for submitting your manuscript to PLOS ONE. After careful consideration, we feel that it has merit but does not fully meet PLOS ONE’s publication criteria as it currently stands. Therefore, we invite you to submit a revised version of the manuscript that addresses the points raised during the review process.

There are a few additional items to consider. First, in addition to responding to the comments from the peer reviewers, please check the writing for clarity and precision. Do not use descriptors unless they are both true and relevant. For example, if truck driving is the most common vocation (and not simply a common vocation) please cite the evidence for this, and if the population of truck drivers is aging, please cite the source for this and explain its importance. As another example, at the end of the first paragraph, you state 'This is nearly 1.4 times that of the general population.' and it would be better to specify 'This is a rate of obesity nearly 1.4 times that of the general (male?) population (in the same age range?).' These are examples from the first paragraph but there are other opportunities to clarify the language throughout the manuscript. Second, was the primary objective of your review to 'identify and describe interventions' but not to assess their effectiveness? This sounds like it might be a scoping review or an overview rather than an evaluation of the available evidence. Please clarify your objective here.

We look forward to receiving your revised manuscript.

Kind regards,

Lisa Susan Wieland

Academic Editor

PLOS ONE

Journal Requirements:

[RI, EP, CvV

National Health and Medical Research Council https://www.nhmrc.gov.au/  grant number GNT1169395

The Transport Workers Union  https://www.twu.com.au/

Linfox https://www.linfox.com/

Centre for Work Health and Safety https://www.centreforwhs.nsw.gov.au/

The funders had no role in study design, data collection and analysis, decision to publish, or preparation of the manuscript.]. 

We note that you received funding from a commercial source: Linfox

Reviewers' comments:

Reviewer's Responses to Questions

**Comments to the Author**

1. Is the manuscript technically sound, and do the data support the conclusions?

Reviewer #1: Yes

Reviewer #2: Partly

2. Has the statistical analysis been performed appropriately and rigorously? 

Reviewer #1: No

Reviewer #2: Yes

3. Have the authors made all data underlying the findings in their manuscript fully available?

Reviewer #1: Yes

Reviewer #2: Yes

4. Is the manuscript presented in an intelligible fashion and written in standard English?

Reviewer #1: Yes

Reviewer #2: Yes

5. Review Comments to the Author

Reviewer #1: 1. Summary of the research and overall impression

Thank you for the opportunity to comment on this systematic review. The paper is well-written and addresses an issue that is undoubtedly of importance. The focus is specifically on weight loss but I think it would be very beneficial if the systematic review also considered outcomes such as type 2 diabetes and cardiovascular disease. Even if there are no data, or very limited data, for those outcomes it would still be a finding in itself to be able to say that these aspects are missing from the current evidence base.

Another aspect that I believe would add value to the systematic review is to attempt some kind of meaningful synthesis. With a little bit of work I think this paper can give the reader a much clearer idea of what the identified studies tell us, where the evidence gaps are and what should be done next in this research area.

2. Discussion of specific areas for improvement

2.1 Major issues

Evidence synthesis: I appreciate that the studies are heterogeneous and not suitable for combining in a traditional meta-analysis but I think the evidence could be synthesised in a more useful way. I would suggest that the authors consider the methods outlined in the recently-published reporting guideline on synthesis without meta-analysis (SWiM) (https://www.bmj.com/content/368/bmj.l6890). Presenting the findings using alternative synthesis methods would really help the reader get a better sense of what the evidence says (or indeed what is lacking from the evidence). As long as the authors clearly describe their methods when it comes to SWiM, and acknowledge the limitations of this kind of synthesis, I think it would add value to the systematic review.

Interpretation of the evidence: I think the statements in the abstract (“The overall level of evidence in this topic is weak”) and in the conclusion (“the evidence in this area is thin and of questionable quality”) should be clarified substantially. I imagine the authors are referring to certainty of evidence but it is not clear if they followed any particular process to assess the certainty of the body of evidence. I think it would be useful to use the GRADE approach of assessing the certainty of evidence. This would also involve producing a Summary of Findings table showing the level of certainty of evidence for each important outcome.

Table 1: Have the authors considered using this table to present only the characteristics of the studies and then presenting the outcome data separately? I think Table 1 should be used to demonstrate the similarities and differences between the studies in terms of design, participants and interventions. A separate table or other format of presentation for the outcome data on weight (and other important outcomes, e.g. T2DM) will then be easier to follow for the reader. If the outcome data are presented separately as suggested, please consider converting all weight data to kg rather than presenting some data in lbs and some in kg. Furthermore, rather than stating ‘not significant’ to describe differences between groups, please present the p-value or other similar statistic so that the reader has all the pertinent information.

Conclusion: please consider specifying in more detail about what form future research should take in this area, e.g. what would be ideal RCT look like? What comparisons would be most useful? What components of MI and supporting resources should be investigated? What characteristics should the participants have, e.g. should they be any transport workers or specifically truck drivers, should they be limited to those that have obesity? How long should the participants be followed up for?

2.2 Minor issues

Search strategy: I would like to see a slightly more comprehensive search strategy that includes grey literature and studies published in languages other than English.

Inclusion criteria and title: Can the authors clarify whether the included studies only recruited participants who had obesity? The title of the article refers specifically to truck drivers with obesity but the inclusion criteria as described in the abstract and methods section suggest that studies with truck drivers of any weight, not just truck drivers with obesity, were eligible. If that is the case then perhaps the title should be simply ‘A systematic review of weight loss interventions in truck drivers’, i.e., regardless of their weight at baseline.

Risk of bias assessment: can the authors clarify how they used the ROB2 tool? Since ROB2 is an outcome-based tool, not study-based, it should be made clear that the risk of bias was assessed for particular outcomes rather than for each study. Did they assess risk of bias in terms of effect of assignment to the intervention or the effect of adhering to the intervention? Additionally, I think it is important not to refer to risk of bias as a synonym for study quality; the Cochrane risk of bias tool is designed only to assess risk of bias, it does not claim to assess study quality.

Interaction between smoking cessation and weight loss: there is nothing mentioned in Methods about smoking but in the results there is a section on smoking cessation and weight loss. Were truck drivers who smoke a particular subgroup of interest (either specified in advance or identified during the process of the review)? I found it quite difficult to follow what was done in this trial. It seems to be an intervention focused on weight loss and smoking cessation at the same time, so I would expect all the participants to be smokers, but it also seems that there were non-smokers in the trial. Please can the authors clarify what the intervention was and what characteristics the participants had? Perhaps there is scope to explore the effect of a weight loss intervention in non-smokers compared to those who are trying to give up (but this would be just a hypothesis-generating subgroup rather than providing any robust evidence).

Strengths and limitations: If the authors decide to use the SWiM guideline, they can use this section to highlight what the SWiM approach adds to the paper as well as outlining the limitations of SWiM.

General comments: I think it is better to avoid the wording ‘obese drivers’ and use ‘drivers with obesity’ instead. I would also suggest replacing generalisations such as ‘unhealthy food’ with something like ‘diet high in saturated fat’.

Reviewer #2: The effect of weight loss interventions in truck drivers with obesity: Systematic review

Truck driving is the most common vocation among males and associated with high levels of overweight/obesity and unhealthy lifestyles. The current study is a systematic review which aimed to identify and discuss the quality of intervention studies addressing weight loss in male truck drivers. The level of evidence presented overall by the review regarding interventions targeting this domain was weak. The study is important and, in my view, provides a useful contribution to the field. However, there are a number of areas that would benefit from further clarification before the manuscript is suitable for publication. I have provided comments below outlining my suggestions. I hope these comments are helpful to the author(s).

General comment – Inclusion of both line and page numbers would substantially facilitate the peer review process.

Introduction

Page 4: ‘This aging and predominately male dominated population’ – Please provide some further context to describe the population such as mean/average age or male truck drivers.

Page 4: ‘Approximately 80% of these participants also failed to meet the National guidelines and recommendations for physical activity per week’ - Please provide further evidence outlining the independent (i.e. in addition to overweight/obesity) health risks of physical inactivity and sedentary behaviour (that is not meeting physical activity for health guidelines and spending too much time sitting).

Page 4: ‘That cross-sectional survey found that 63% of the truck drivers consumed at least one serving of unhealthy food each day’ – Please provide more details from the study i.e. define what was defined specifically as ‘unhealthy food’.

Page 4 (and generally): Physical activity and exercise are used to define related but distinct constructs. That is, according to Caspersen and colleagues ‘Physical activity is defined as any bodily movement produced by skeletal muscles that results in energy expenditure … Exercise is a subset of physical activity that is planned, structured, and repetitive and has as a final or an intermediate objective the improvement or maintenance of physical fitness’ (see - Caspersen, C. J., Powell, K. E., & Christenson, G. M. (1985). Physical activity, exercise, and physical fitness: definitions and distinctions for health-related research. Public health reports (Washington, D.C. : 1974), 100(2), 126–131.). I suggest referring to the broader concept of physical activity (as is the norm in the field of physical activity-related research) unless specifically referring to exercise (as defined above).

Page 5: ‘A meta-analysis of 45 studies investigating weight loss interventions focusing on both food intake and physical activity with obese adults (17) showed that participants with effective and sustained behavioral strategies were less likely to regain their lost weight’ – Please outline what these specific behavioural modifications or behaviour change techniques (BCTs) were? Other similar reviews have employed novel/robust methods (i.e. meta-regression) to identify intervention components or BCTs linked to more successful weight loss interventions, including provision of instructions, self-monitoring, relapse prevention and prompting practice (i.e. see - Stephan U. Dombrowski, Falko F. Sniehotta, Alison Avenell, Marie Johnston, Graeme MacLennan & Vera Araújo-Soares (2012) Identifying active ingredients in complex behavioural interventions for obese adults with obesity-related co-morbidities or additional risk factors for co-morbidities: a systematic review, Health Psychology Review, 6:1, 7-32, DOI: 10.1080/17437199.2010.513298). Moreover, studies including more BCTs aimed at dietary change and aligned with with Control Theory have been associated with greater weight loss. These are important factors to recognise specifically when conducting systematic reviews (i.e. key strength of meta-regression enables identification of intervention ‘active ingredients’ or BCTs more likely to be effective).

Methods

Page 6: In addition to the type of intervention (format e.g. group-based/community or individual level (i.e. one-to-one) or mode of delivery (online or face-to-face etc)) was there consideration given to the context of the intervention (e.g. country-specific/systems including education, healthcare etc)?

Page 6: ‘Data were extracted independently onto an excel spread sheet by two reviewers to identify the interventions, population groups, and outcomes of each study’ – Please confirm if this relates to the spreadsheet included as a supplementary file. If so, please include link/reference here.

Results

Page 8: ‘Details of included studies and interventions’ - Due to the heterogeneity of interventions described/delivered it is difficult to follow this section. Would it be possible to further specify the type and format of the interventions/studies into different types (e.g. individual, group-based or online etc)? It would also be illuminating to discern intervention content, including evidence-based behaviour change techniques described within the intervention protocols.

Page 10: ‘Preliminary findings for weight reduction and BMI’ – Please describe in more detail how weight (outcomes) was assessed (objectively) e.g. as percentage weight loss (e.g. five to 10 percent body weight), waist circumference (cm or inches) or average weight (kilograms or pounds) etc.

Discussion

Page 12: The focus comes across as being disproportionately focused on MI and insufficient attention currently given to additional intervention components (or BCTs) potentially associated with greater intervention efficacy for this target group.

Page 12: ‘As truck drivers are on the road for many hours each day, the study supports the possibility of further trials using MI coaching with telehealth delivery, to positively impact weight and health taking behaviors’ – Please articulate further how this could be delivered and/or what means could be employed to reduce associated inequalities (e.g. access to technologies).

Page 12: Weight loss maintenance has been identified as being a particularly challenging area of weight-related intervention research. Please describe the exact time scales of follow-up time points included in the studies within this review (e.g. 12 months) and what methods could be employed in future to enhance follow-up assessments and to support weight maintenance (from the evidence presented).

Pages 12 & 13: ‘MI is often considered the best coaching approach for health behaviour change’ - It is important to link to other potential theoretical approaches within weight loss literature. For example, there has been some discussion of potential overlap regarding theoretical constructs from MI with other prominent theories of behaviour change, especially in relation to weight loss maintenance, such as Self-Determination Theory (e.g. autonomous motivation). This is particularly important when considering the importance of (adaptive) coaching style and motivational climate.

Page 13: ‘Along with the small numbers in many of the studies’ – Small numbers of what? Sample size? Please be more specific.

Page 13: Please consider the importance of gender sensitivity, especially regarding the context, content/style of interventions included in the review, specifically regarding future research in the field. Research involving professional sport settings, congruent with masculine norms have shown to be potent ways of encouraging men to participate in weight-loss programmes in other domains within Australia, UK and other countries (e.g. Kwasnicka D, Ntoumanis N, Hunt K, Gray CM, Newton RU, et al. (2020) A gender-sensitised weight-loss and healthy living program for men with overweight and obesity in Australian Football League settings (Aussie-FIT): A pilot randomised controlled trial. PLOS Medicine 17(8): e1003136. https://doi.org/10.1371/journal.pmed.1003136), thus may be applicable/transferable to males in occupations including truck drivers.

6. PLOS authors have the option to publish the peer review history of their article (what does this mean?). If published, this will include your full peer review and any attached files.

Reviewer #1: **Yes: **Fiona Stewart

Reviewer #2: No

---

## [Author Response · Author response to Decision Letter 0]

14 Jun 2021

19.5.21

Dear Lisa Susan Wieland and reviewers,

Thank you for the opportunity to respond to the reviewer concerns and resubmit our manuscript: The effect of weight loss interventions in truck drivers: Systematic review

In the table below we have presented the reviewer concerns (column 1), our response and the place in the manuscript where any changes can be found (columns 2 and 3). We hope these changes meet your recommendations.

Kind regards

Editor requests 

Do not use descriptors unless they are both true and relevant. For example, if truck driving is the most common vocation (and not simply a common vocation) please cite the evidence for this, Evidence has been added to the first sentence of the intro. “Truck driving is the most common occupation among males in 29/50 of the United States (1) and also across Australia (2) with an estimated number of drivers as 3.2 million in Europe (2015) (3)…”

if the population of truck drivers is aging, please cite the source for this and explain its importance. This has been added into the intro, paragraph one “In the US >85% of truck drivers are male with a median age of 46 compared with national average of workers at 41 years adding additional health risks and issues around attrition of workforce as they age (8). In Australia, the average age of truck drivers is 44.3 compared to the average worker at 39 years (9) with 6.5% reported as female (10). ..”

at the end of the first paragraph, you state ‘This is nearly 1.4 times that of the general population.' and it would be better to specify 'This is a rate of obesity nearly 1.4 times that of the general (male?) population (in the same age range?).' This has been added to first paragraph in introduction. This is a rate of obesity that is nearly 1.4 times that of the general male population across similar age groups (16).

was the primary objective of your review to 'identify and describe interventions' but not to assess their effectiveness This has been adjusted in the aims – at end of the intro. “Therefore, the primary objective of this systematic review was to explore interventions for weight reduction in truck drivers and identify which interventions are effective for weight reduction in truck drivers to inform a future pilot program in Australia.”

Reviewer request Response Description/quote

(Line numbers are correct for the tracked changes version)

R 1:

Thank you for the opportunity to comment on this systematic review. The paper is well-written and addresses an issue that is undoubtedly of importance. The focus is specifically on weight loss but I think it would be very beneficial if the systematic review also considered outcomes such as type 2 diabetes and cardiovascular disease. Even if there are no data, or very limited data, for those outcomes it would still be a finding in itself to be able to say that these aspects are missing from the current evidence base.

 Thank you for your query. We decided upon this search as the results were to inform an intervention pilot in the area of addressing weight/loss in truck drivers.

If we were to broaden the outcomes to other areas of chronic disease, we would need to re-execute several stages of the review, which would ultimately then fall outside the scope of this project. 

We have added into the limitations the following statement to clarify this narrow scope of the project. 

 Line 398-401: 

“One of the limitations is the narrow scope of this review exploring interventions to effect weight loss and not considering other chronic conditions e.g. Diabetes, Cardiovascular disease. Although these outcomes are very much interconnected in health, the scope of the review was determined so as to inform a potential intervention pilot focusing on truck driver weight loss.”

Another aspect that I believe would add value to the systematic review is to attempt some kind of meaningful synthesis. With a little bit of work I think this paper can give the reader a much clearer idea of what the identified studies tell us, where the evidence gaps are and what should be done next in this research area. The synthesis now follows more of the Swim structure and is easier to follow and provides more detail around the findings of the outcomes. 

Research gaps have been more clearly defined in several sections including discussion (lines 372-380) and in the limitations section. Lines 382 - 390

“These studies highlight the research gaps that still exist in the area of effective weight reduction programs for truck drivers. Well conducted clinical trials in larger sample groups are required to produce high-level evidence of effective interventions to reduce driver weight, applicability of interventions to the gender sensitivities of a predominantly male driver population, and consistency of measurement of outcomes to enable comparison across studies in the future. Ongoing research needs to focus on addressing these gaps to ameliorate the ongoing negative health and wellbeing implications of obesity for truck drivers.

R 1: Evidence synthesis: I appreciate that the studies are heterogeneous and not suitable for combining in a traditional meta-analysis but I think the evidence could be synthesised in a more useful way.

I would suggest that the authors consider the methods outlined in the recently-published reporting guideline on synthesis without meta-analysis (SWiM) (https://www.bmj.com/content/368/bmj.l6890). Presenting the findings using alternative synthesis methods would really help the reader get a better sense of what the evidence says (or indeed what is lacking from the evidence). As long as the authors clearly describe their methods when it comes to SWiM, and acknowledge the limitations of this kind of synthesis, I think it would add value to the systematic review.

 Thank you for this suggestion. While we followed this process, this was not clearly identified in the manuscript. 

We have now added details of this framework into the methods and also structured the synthesis this way in the results to give the reader more clarity. Lines: 164 – 178:

“The Synthesis Without Meta-analysis (SWiM) framework was used to guide the synthesis of data (31). Groupings were study design (RCTs and quasi RCT; pre-post studies); outcomes including weight, BMI, fat mass, and measurements; the method and length of intervention delivery; and the target group. Where possible, the standardized metric the effect size (Cohen’s d) of the intervention was reported from the study or calculated where possible, to enable comparison of intervention effects across the different outcome measures applied (32). The d statistic was interpreted as 0.2 representing a small effect, 0.5 a medium effect and 0.8 and higher a large effect (32). For studies where effect size was unable to be calculated p values, median and interquartile range were reported. Criteria used to prioritize the findings were study design (those with a control) and those where risk of bias assessment was either low, moderate or some concerns. Those with high or critical risk of bias concerns is not discussed in detail. Investigations for heterogeneity were not prespecified prior to analysis as the breadth of data was not yet determined in this area. Findings are presented in tables (key characteristics and outcomes) and figures (risk of bias findings). A description of the synthesis of findings is presented and related to answering the research question. Limitations of the study and synthesis is also reported.”

R 1:

Interpretation of the evidence: I think the statements in the abstract (“The overall level of evidence in this topic is weak”) and in the conclusion (“the evidence in this area is thin and of questionable quality”) should be clarified substantially. I imagine the authors are referring to certainty of evidence but it is not clear if they followed any particular process to assess the certainty of the body of evidence. 

I think it would be useful to use the GRADE approach of assessing the certainty of evidence. This would also involve producing a Summary of Findings table showing the level of certainty of evidence for each important outcome.

 Thank you for the suggestion of using the GRADE approach for the as a framework for the overall RoB discussion, and we would usually follow this approach for systematic reviews that explore RCTs only.

We set out to find the highest level of evidence in this area to inform the intervention and identified the lack of research in this area.

We explored the possibility of completing a GRADE summary of findings table. However, as there were only two papers included that were RCTs, and only two common outcomes across these studies, we believe that adding the GRADE summary would add a layer of complexity that is not required to clearly describe the body of available evidence. 

Our review shows there is a low level of evidence (only two RCTs) to address the research question, and the bias present in the two RCTs is not a strong basis on which to make clinical decisions. 

Therefore, we have not included a GRADE approach examining outcomes in the revisions. Rather, we have clarified our statements regarding the body of evidence in the abstract and the conclusion.

 The abstract now reads, Lines 43, 44:

“Based on the small number of RCTs and the biases they contain, the overall level of evidence in this topic is weak.”

Lines – 407 – 409

The conclusion now reads, “However, the level of evidence in this area is minimal with only two RCTs available. Findings presented are also inconclusive due to the level of bias, small sample sizes, and designs of each of the studies included in this review.”

R1:

Table 1: Have the authors considered using this table to present only the characteristics of the studies and then presenting the outcome data separately? I think Table 1 should be used to demonstrate the similarities and differences between the studies in terms of design, participants and interventions. 

A separate table or other format of presentation for the outcome data on weight (and other important outcomes, e.g. T2DM) will then be easier to follow for the reader. 

 Thank you for your suggestion. We toyed with both options at the time of writing this paper. 

We have now separated the characteristics and outcomes as suggested. Table 2

If the outcome data are presented separately as suggested, please consider converting all weight data to kg rather than presenting some data in lbs and some in kg. 

 Converted all weights into Kg as requested – Table 1 

Furthermore, rather than stating ‘not significant’ to describe differences between groups, please present the p-value or other similar statistic so that the reader has all the pertinent information.

 This section has been removed as was relating to smoking outcomes which is no longer included. The information stated in table 2 (outcomes) are stated as reported in the papers included in the review. 

Conclusion: please consider specifying in more detail about what form future research should take in this area, e.g. what would be ideal RCT look like? 

What comparisons would be most useful? 

What components of MI and supporting resources should be investigated? 

.

What characteristics should the participants have, e.g. should they be any transport workers or specifically truck drivers, should they be limited to those that have obesity? 

How long should the participants be followed up for?

 We have added greater detail to the conclusion, however for reasons of space we have kept our recommendations to a high level. We have recommended at least a 12 month follow up and that interventions be described in sufficient detail to allow replication in other studies, amongst other key concepts. Lines 406 – 415:

The conclusion now reads: 

“A combination of MI and supporting resources has potential for long-term effectiveness in reducing truck driver body weight. However, the level of evidence in this area is minimal with only two RCTs available. Findings presented are also inconclusive due to the level of bias, small sample sizes, and designs of each of the studies included in this review. Further clinical trials are required with larger cohorts of truck drivers, and need to aim to include 12-month or longer follow-up periods, provide clear and detailed description of the intervention so they can be replicated elsewhere, use consistent measurement of weight reduction outcomes, and administer evidence-based interventions appropriate for the gender sensitivities within the industry. Future studies could then determine what interventions can be transferred to the Australian transport industry for weight reduction and how they could be sustainably implemented.” 

Search strategy: I would like to see a slightly more comprehensive search strategy that includes grey literature and studies published in languages other than English.

 While it would have been great to include grey literature and all articles of any language in this review, we were not able for the following reasons:

• The review needed to provide peer reviewed evidence to inform an intervention

• We had restricted resources for the review during the 2020 pandemic

• We did not have access to interpretation services for papers

Clarification has been added into the inclusion criteria. Line 123

Inclusion criteria were peer reviewed experimental and quasi-experimental primary…

Line 124/125.

Studies published in English (as translation resources were unavailable) in…

Inclusion criteria and title: 

Can the authors clarify whether the included studies only recruited participants who had obesity? 

The title of the article refers specifically to truck drivers with obesity but the inclusion criteria as described in the abstract and methods section suggest that studies with truck drivers of any weight, not just truck drivers with obesity, were eligible. 

If that is the case then perhaps the title should be simply ‘A systematic review of weight loss interventions in truck drivers’, i.e., regardless of their weight at baseline. Have changed the title as suggested, to: 

The effect of weight loss interventions in truck drivers: Systematic review

Risk of bias assessment: 

can the authors clarify how they used the ROB2 tool? 

Since ROB2 is an outcome-based tool, not study-based, it should be made clear that the risk of bias was assessed for particular outcomes rather than for each study. 

 Additional information has been included to show how we have adhered to the Cochrane guidelines for use of the RoB2 tool.

 Lines 143 – 154

“The effect of assignment regarding was the following outcomes (weight, BMI, fat mass, body measurements), converting all to kilograms (kg) and centimeters (cm) for ease of comparison, reporting effect sizes between baseline and reassessment and/or follow-up if provided. All assessments were completed independently by two people (CK, EP or CK, NN). RoB 2 has five criteria of bias: 1. Arising from the randomization process; 2. Due to deviations from intended interventions; 3. Due to missing outcome data; 4. In measurement of the outcome; and 5. In selection of the reported result. Each bias criterion is rated as either low risk, some concerns or high risk when using the tool logarithms and determined by these criteria. The Microsoft Excel RoB 2 tool from Cochrane was used for each study, as there were only two outcomes that were common across two studies (30). All eligible studies are discussed in this review regardless of their RoB results as there is limited work in this area. Sub-group analysis is presented where possible.”

Did they assess risk of bias in terms of effect of assignment to the intervention or the effect of adhering to the intervention? 

 Stated in the methods section Lines 133 – 144

 (as above) 

Additionally, I think it is important not to refer to risk of bias as a synonym for study quality; the Cochrane risk of bias tool is designed only to assess risk of bias, it does not claim to assess study quality.

 References to quality of the article have been removed when talking about risk of bias. Lines: 138 (heading – Risk of Bias Assessment 

Line 139, 148, 156, 350, 353, 370

Interaction between smoking cessation and weight loss: there is nothing mentioned in Methods about smoking but in the results there is a section on smoking cessation and weight loss. 

Were truck drivers who smoke a particular subgroup of interest (either specified in advance or identified during the process of the review)? 

I found it quite difficult to follow what was done in this trial. It seems to be an intervention focused on weight loss and smoking cessation at the same time, so I would expect all the participants to be smokers, but it also seems that there were non-smokers in the trial. Please can the authors clarify what the intervention was and what characteristics the participants had? 

Perhaps there is scope to explore the effect of a weight loss intervention in non-smokers compared to those who are trying to give up (but this would be just a hypothesis-generating subgroup rather than providing any robust evidence).

 Agree that this tends to cloud the findings. Information on smoking has been removed. Lines 288-295 

Strengths and limitations: If the authors decide to use the SWiM guideline, they can use this section to highlight what the SWiM approach adds to the paper as well as outlining the limitations of SWiM. Thank you for identifying the need to state the components of the SWiM framework.

This has been done and the strengths/limitations of using the SWiM have now been included as well. Lines 392 – 401:

“The strengths of this review are the specificity of the inclusion criteria, the number of databases explored, rigor in which the screening, quality RoB assessment and data extraction were conducted, (reducing the risk of selection bias for this review) and the use of the structured approach to the analysis and synthesis without meta-analysis (SWiM). The SWiM framework allowed for a clearer description of the methods, provided clarity of the links of the synthesis and a checklist for how to group and report the findings. There were also no conflicts of interest for the authors.

One of the limitations is the narrow scope of this review exploring interventions to effect weight loss and not considering other chronic conditions e.g. Diabetes, Cardiovascular disease. Although these outcomes are very much interconnected in health, the scope of the review was determined so as to inform a potential intervention pilot focusing on truck driver weight loss.”

I think it is better to avoid the wording ‘obese drivers’ and use ‘drivers with obesity’ instead. 

 Changed in lines 67 - 69 This includes crashes, where drivers with obesity were reported as being twice as likely to crash than drivers in the normal weight range (16).

I would also suggest replacing generalisations such as ‘unhealthy food’ with something like ‘diet high in saturated fat’. This was stated as per the study cited, as ‘unhealthy’ with an explanation of what this means in line 68. This has now been clarified by adding ‘ ‘ around the word. Lines 82 – 84 :

“identified as ‘unhealthy’ with high fat and carbohydrate content (13). The cross-sectional survey found that 63% of the truck drivers consumed at least one serving of ‘unhealthy’ food each day (13).”

Inclusion of both line and page numbers Done as requested. 

P 4 This aging and predominately male dominated population’ – 

Please provide some further context to describe the population such as mean/average age or male truck drivers.

 Additional information has now been added regarding this. Lines 64 – 67

“In the US >85% of truck drivers are male with a median age of 46 compared with national average of workers at 41 years (6). In Australia, the average age of truck drivers is 44.3 compared to the average worker at 39 years (7) with 6.5% reported as female (8).”

‘Approximately 80% of these participants also failed to meet the National guidelines and recommendations for physical activity per week’ – 

Please provide further evidence outlining the independent (i.e. in addition to overweight/obesity) health risks of physical inactivity and sedentary behaviour (that is not meeting physical activity for health guidelines and spending too much time sitting).

 This has been addressed with the following sentence. Lines 85 – 86:

“The health risks of sedentary occupations are well documented and include cardiovascular disease, cardiorespiratory problems, obesity, diabetes and even cancer which creates cyclical health risks (18).”

Page 4: ‘That cross-sectional survey found that 63% of the truck drivers consumed at least one serving of unhealthy food each day’ – 

Please provide more details from the study i.e. define what was defined specifically as ‘unhealthy food’. Changed in line 72

Also added info to line 81 …at least one serving of food high in fats and carbohydrates each day…

… through healthy food choices (low in sugars and fats) and…

Physical activity and exercise are used to define related but distinct constructs. That is, according to Caspersen and colleagues ‘Physical activity is defined as any bodily movement produced by skeletal muscles that results in energy expenditure … Exercise is a subset of physical activity that is planned, structured, and repetitive and has as a final or an intermediate objective the improvement or maintenance of physical fitness’ (see - Caspersen, C. J., Powell, K. E., & Christenson, G. M. (1985). Physical activity, exercise, and physical fitness: definitions and distinctions for health-related research. Public health reports (Washington, D.C. : 1974), 100(2), 126–131.). 

I suggest referring to the broader concept of physical activity (as is the norm in the field of physical activity-related research) unless specifically referring to exercise (as defined above). This has been changed for the areas where this is relevant and where studies refer to exercise equipment or facilities in relation to the interventions, this has been left. Lines 42, 90, 91, 93, 128, 244, 245, 312, 321, 

Page 5: ‘A meta-analysis of 45 studies investigating weight loss interventions focusing on both food intake and physical activity with obese adults (17) showed that participants with effective and sustained behavioral strategies were less likely to regain their lost weight’ – 

Please outline what these specific behavioural modifications or behaviour change techniques (BCTs) were? Other similar reviews have employed novel/robust methods (i.e. meta-regression) to identify intervention components or BCTs linked to more successful weight loss interventions, including provision of instructions, self-monitoring, relapse prevention and prompting practice (i.e. see - Stephan U. Dombrowski, Falko F. Sniehotta, Alison Avenell, Marie Johnston, Graeme MacLennan & Vera Araújo-Soares (2012)

 Identifying active ingredients in complex behavioural interventions for obese adults with obesity-related co-morbidities or additional risk factors for co-morbidities: a systematic review, Health Psychology Review, 6:1, 7-32, DOI: 10.1080/17437199.2010.513298). Moreover, studies including more BCTs aimed at dietary change and aligned with with Control Theory have been associated with greater weight loss. 

These are important factors to recognise specifically when conducting systematic reviews (i.e. key strength of meta-regression enables identification of intervention ‘active ingredients’ or BCTs more likely to be effective).

 These have been outlined as suggested. Lines 335 – 343

“MI is often considered an effective coaching approach for health behavior change (41, 45) and was the predominant approach described in 4/6 of the studies that used counselling (4, 24, 36, 38). It is interesting to note that there are additional theoretical behavior change approaches described in weight loss literature which include (but not limited to) Self-determination Theory (46) (effective in maintaining weight loss) (47); Social Cognitive Therapy, Transtheoretical Model, and Theory of Planned Behavior, all of which have been used to manage and maintain weight loss in adults (48). Additional to the four studies that used a MI approach the other two used ‘counselling’ without stating their behavior change approach and therefore MI is explored in more detail.”

Methods: Page 6: In addition to the type of intervention (format e.g. group-based/community or individual level (i.e. one-to-one) or mode of delivery (online or face-to-face etc)) was there consideration given to the context of the intervention (e.g. country-specific/systems including education, healthcare etc)?

 This information h as been added to Table 1 (Type of intervention column) – where it was available in the papers.

Narrative has also been added to clarify this more. Lines 232-241

Interventions were all targeted at the individual driver with 6/7 providing one-on-one sessions via the phone (4, 20, 27-30); with two providing face-to-face sessions one at a health clinic (30) and one stated the counsellors traveled to the drivers (26); two provided self-paced online training (4, 20); one provided cooking and exercise equipment (29); four provided health messaging and resources (28-31); and two using the SHIFT program included a team competition for weight loss and driver safety (4, 20).

Page 6: ‘Data were extracted independently onto an excel spread sheet by two reviewers to identify the interventions, population groups, and outcomes of each study’ – 

Please confirm if this relates to the spreadsheet included as a supplementary file. If so, please include link/reference here. Yes, this is the supplementary file – added in this reference to this file. Line 136

Results

Page 8: ‘Details of included studies and interventions’ - Due to the heterogeneity of interventions described/delivered it is difficult to follow this section. 

Would it be possible to further specify the type and format of the interventions/studies into different types (e.g. individual, group-based or online etc)? 

 A column has now been added into the characteristics table A1 Column heading: Type of intervention delivery

It would also be illuminating to discern intervention content, including evidence-based behaviour change techniques described within the intervention protocols.

 Where possible and stated in the studies included in the review, the intervention delivery methods have been added to table 1. Intervention content was included in the intervention type column of Table A1.

Any gaps are due to the information not being reported.

We have also added in additional sentences to cover this in the results. Table 1 column 5

Lines 232 - 241:

Interventions were all targeted at the individual driver with 6/7 providing one-on-one sessions via the phone (4, 24, 35-38); with two providing face-to-face sessions one at a health clinic (38) and one stated the counsellors traveled to the drivers (34); two provided self-paced online training (4, 24); one provided cooking and exercise equipment (37); four provided health messaging and resources (36-39); and two used the Safety and Health Involvement For Truckers (SHIFT) program which included a team competition for weight loss and driver safety (4, 24). Sendall et al. (2016) tested health promotion interventions (no coaching), to identify how these impacted the knowledge and behavior of drivers. The findings had varied results across the reported levels of obese or overweight drivers, however, only an overall reduction in self-reported BMI from the obese participants was identified (39).

Page 10: ‘Preliminary findings for weight reduction and BMI’ – 

Please describe in more detail how weight (outcomes) was assessed (objectively) e.g. as percentage weight loss (e.g. five to 10 percent body weight), waist circumference (cm or inches) or average weight (kilograms or pounds) etc.

 All outcome results (including weights, measurements and effect sizes) are reported as described in the study results sections for those included in the review.

 Line 141 - 144

The effect of assignment will be used regarding the following outcomes (weight, BMI, fat mass, body measurements), converting all to kilograms and centimeters for ease of comparison, reporting effect sizes between baseline and reassessment and/or follow-up if provided. All assessments were completed independently by two people (CK, EP or CK, NN).

Discussion

Page 12: The focus comes across as being disproportionately focused on MI and insufficient attention currently given to additional intervention components (or BCTs) potentially associated with greater intervention efficacy for this target group.

 This has been addressed by adding the following sentences. Lines, 318-325.

 MI is often considered an effective coaching approach for health behavior change (41, 45) and was the predominant approach described in 4/6 of the studies that used counselling (4, 24, 36, 38). It is interesting to note that there are additional theoretical behavior change approaches described in weight loss literature which include (but not limited to) Self-determination Theory (46) (effective in maintaining weight loss) (47); Social Cognitive Therapy, Transtheoretical Model, and Theory of Planned Behavior, all of which have been used to manage and maintain weight loss in adults (48). Additional to the four studies that used a MI approach the other two used ‘counselling’ without stating their behavior change approach and therefore MI is explored in more detail. 

Page 12: ‘As truck drivers are on the road for many hours each day, the study supports the possibility of further trials using MI coaching with telehealth delivery, to positively impact weight and health taking behaviors’ – 

Please articulate further how this could be delivered and/or what means could be employed to reduce associated inequalities (e.g. access to technologies). We have added additional information to address this point. Lines 336 – 338:

Careful consideration of the challenges and potential inequalities in providing interventions for this group, such as possibilities of telehealth interventions, making sure the technology for any intervention is accessible and will work when they are travelling and remote, is required.

Page 12: Weight loss maintenance has been identified as being a particularly challenging area of weight-related intervention research. 

Please describe the exact time scales of follow-up time points included in the studies within this review (e.g. 12 months) and what methods could be employed in future to enhance follow-up assessments and to support weight maintenance (from the evidence presented). These stats are included in Table A1 and two of them are discussed in the lines 217 and 221.

However, we have also added this information to the narrative in this section to add more clarity. Line 209:

Of the three studies with a follow-up period across 30-months, 24-months and 6-months respectively…

Pages 12 & 13: ‘MI is often considered the best coaching approach for health behaviour change’ – 

It is important to link to other potential theoretical approaches within weight loss literature. For example, there has been some discussion of potential overlap regarding theoretical constructs from MI with other prominent theories of behaviour change, especially in relation to weight loss maintenance, such as Self-Determination Theory (e.g. autonomous motivation). This is particularly important when considering the importance of (adaptive) coaching style and motivational climate.

 Thank you for your comment. This has been extended upon within the discussion 

 Lines 318 - 325

 MI is often considered an effective coaching approach for health behavior change (41, 45) and was the predominant approach described in 4/6 of the studies that used counselling (4, 24, 36, 38). It is interesting to note that there are additional theoretical behavior change approaches described in weight loss literature which include (but not limited to) Self-determination Theory (46) (effective in maintaining weight loss) (47); Social Cognitive Therapy, Transtheoretical Model, and Theory of Planned Behavior, all of which have been used to manage and maintain weight loss in adults (48). Additional to the four studies that used a MI approach the other two used ‘counselling’ without stating their behavior change approach and therefore MI is explored in more detail. 

Page 13: ‘Along with the small numbers in many of the studies’ – 

Small numbers of what? Sample size? Please be more specific. Line 373 changed to clarify with the small sample sizes in many of the…

Page 13: Please consider the importance of gender sensitivity, especially regarding the context, content/style of interventions included in the review, specifically regarding future research in the field. Research involving professional sport settings, congruent with masculine norms have shown to be potent ways of encouraging men to participate in weight-loss programmes in other domains within Australia, UK and other countries (e.g. Kwasnicka D, Ntoumanis N, Hunt K, Gray CM, Newton RU, et al. (2020) A gender-sensitised weight-loss and healthy living program for men with overweight and obesity in Australian Football League settings (Aussie-FIT): A pilot randomised controlled trial. PLOS Medicine 17(8): e1003136. https://doi.org/10.1371/journal.pmed.1003136), thus may be applicable/transferable to males in occupations including truck drivers. This has been added in the discussion. Lines 345 - 350

Although gender sensitivities were not explored in the reviewed studies, this is another area to investigate in future research. These approaches may provide the way forward for truck drivers (both male and female). Parallels between other male dominated occupations and sport settings, could be explored to identify effective ways of encourage men to participate in weight-loss programs.

---

## [Decision Letter · Decision Letter 1]

8 Jul 2021

PONE-D-21-02019R1

The effect of weight loss interventions in truck drivers: Systematic review

PLOS ONE

Dear Dr. Pritchard,

Thank you for submitting your manuscript to PLOS ONE. After careful consideration, we feel that it has merit but does not fully meet PLOS ONE’s publication criteria as it currently stands. Therefore, we invite you to submit a revised version of the manuscript that addresses the points raised during the review process.

Please consider the additional comments and provide an explanation and rationale if you are unable to address them fully.

We look forward to receiving your revised manuscript.

Kind regards,

Lisa Susan Wieland

Academic Editor

PLOS ONE

Journal Requirements:

Reviewers' comments:

Reviewer's Responses to Questions

**Comments to the Author**

1. If the authors have adequately addressed your comments raised in a previous round of review and you feel that this manuscript is now acceptable for publication, you may indicate that here to bypass the “Comments to the Author” section, enter your conflict of interest statement in the “Confidential to Editor” section, and submit your "Accept" recommendation.

Reviewer #1: (No Response)

Reviewer #2: All comments have been addressed

2. Is the manuscript technically sound, and do the data support the conclusions?

Reviewer #1: Yes

Reviewer #2: Yes

3. Has the statistical analysis been performed appropriately and rigorously? 

Reviewer #1: Yes

Reviewer #2: N/A

4. Have the authors made all data underlying the findings in their manuscript fully available?

Reviewer #1: Yes

Reviewer #2: Yes

5. Is the manuscript presented in an intelligible fashion and written in standard English?

Reviewer #1: Yes

Reviewer #2: Yes

6. Review Comments to the Author

Reviewer #1: Thank you for the opportunity to review this manuscript again. The authors have clearly done a great deal of valuable work and the review is now much improved. I have a few outstanding points to mention:

- My previous comments with regard to other outcomes such as type 2 diabetes may not have been clear enough. My suggestion was to examine the studies already included in the review and report whether or not the participants’ T2DM improved (if they already had T2DM), or if they developed T2DM (or other long-term conditions associated with obesity) during the follow-up period. Since one of the main reasons weight loss is encouraged in people with obesity is to prevent or improve conditions such as T2DM (i.e. weight in itself is largely a surrogate outcome) therefore I think it is important to capture whether or not these interventions have an effect on weight loss and on other outcomes. I can see from the search strategy that you would not need to go back and search all over again, the issue would be to extract more data from the studies that are already in the review.

- The text still has some typos and grammar errors, e.g. subject-verb agreements. There are also still some instances of language that could be perceived as judgmental, e.g. ‘obese truck drivers’ instead of ‘truck drivers with obesity’.

- It would be great to see the findings put into some kind of clinical context. For instance, at the moment the abstract reads as if there were some statistically significant differences found in favour of the interventions but it is not clear if those differences are clinically meaningful. It would be very useful for the reader to see some indication of the clinical significance (or not) in the abstract, results and discussion sections.

- Is there a typo in the following sentence? I am not sure what it is trying to say: “The effect of assignment regarding was the following outcomes (weight, BMI, fat mass, body measurements), converting all to kilograms (kg) and centimeters (cm) for ease of comparison, reporting effect sizes between baseline and reassessment 140 and/or follow-up if provided.”

- Discussion: “This systematic review has identified seven intervention studies conducted over the past 20 years that explored the effect of weight loss interventions on obese truck drivers.” I don’t think the latter part of the sentence is completely accurate since the inclusion criteria did not specify that the truck drivers had to have obesity, rather the review focuses on studies of lifestyle interventions for truck drivers in general, regardless of their starting weight. In other words, I suggest removing the word ‘obese’ from this sentence.

Reviewer #2: The author(s) have sufficiently addressed each of my comments. I have no further comments other than to thank the authors for the opportunity to read and review their work.

7. PLOS authors have the option to publish the peer review history of their article (what does this mean?). If published, this will include your full peer review and any attached files.

Reviewer #1: **Yes: **Fiona Stewart

Reviewer #2: No

---

## [Author Response · Author response to Decision Letter 1]

28 Jul 2021

28 July 2021

Dear Lisa Susan Wieland,

Thank you for this opportunity to hone this manuscript to the next stage for publication. We have addressed each of the comments from Reviewers and highlight our response below.

PLOS One – review comments #2, received 9 July 2021.

 - Reviewer #1: Thank you for the opportunity to review this manuscript again. The authors have clearly done a great deal of valuable work and the review is now much improved. I have a few outstanding points to mention:

- My previous comments with regard to other outcomes such as type 2 diabetes may not have been clear enough. My suggestion was to examine the studies already included in the review and report whether or not the participants’ T2DM improved (if they already had T2DM), or if they developed T2DM (or other long-term conditions associated with obesity) during the follow-up period. Since one of the main reasons weight loss is encouraged in people with obesity is to prevent or improve conditions such as T2DM (i.e. weight in itself is largely a surrogate outcome) therefore I think it is important to capture whether or not these interventions have an effect on weight loss and on other outcomes. I can see from the search strategy that you would not need to go back and search all over again, the issue would be to extract more data from the studies that are already in the review.

Added outcomes in methods – Page 7:… or impact on long-term health conditions associated with obesity e.g. Type-2 diabetes, or Metabolic Syndrome (MeS), that raises the risk of heart disease.

Added paragraph in results: page 10

Four studies explored additional health risk factors relating to weight i.e. two measured blood glucose levels with readings identified as 95.81 (mg/dl) pre and 110.44 post, both of which were in the normal range (not significant at p=0.46) with no follow-up testing done at 30 months [26], and the other reported 123 (mg/dl) at baseline decreasing to 98 on exit (not significant at p=0.79) [39]. One reported the presence of diabetes (type not stated) with control at 13% at baseline, the intervention group at 12% and blood glucose risk with slight increase across both groups at 6 months which was not significant (p=0.84) [6], and one study reported the presence of Type-2 diabetes as 1% at baseline (no follow-up data) with over two-thirds (71%) having MeS at baseline which decreased in the two intervention groups at 12 months to 62% (in the LIFE group) and to 60% (in the REF group), also not significant (p=0.34) [37].

In the discussion: Page 19/20

None of the four studies that measured incidence of diabetes or MeS, showed a significant reduction [6, 26, 37, 39].

- The text still has some typos and grammar errors, e.g. subject-verb agreements. 

Changed the following: 

Page 1 to: ‘dominated population are exposed’

Page 6: ‘interventions were effective for weight’

Page 7: ‘All eligible studies were discussed in this’

Page 8 ‘Findings have been presented’

Page 21: ‘note that there were additional theoretical’

Page 22: ‘which also impacted the findings’

Page 23 ‘The strengths of this review were the specificity’

Page 23: ‘We discovered there were few high-quality’

- There are also still some instances of language that could be perceived as judgmental, e.g. ‘obese truck drivers’ instead of ‘truck drivers with obesity’. 

 (only one was found and removed page 19 as per comment below)

- It would be great to see the findings put into some kind of clinical context. For instance, at the moment the abstract reads as if there were some statistically significant differences found in favour of the interventions but it is not clear if those differences are clinically meaningful. It would be very useful for the reader to see some indication of the clinical significance (or not) in the abstract, results and discussion sections.

Sentence added in the abstract 

Results: The effect sizes for 5/7 studies were medium to large size (5/7 studies), indicating likely clinical significance.

Conclusion: 

Interventions that include a combination of coaching and other resources may provide successful weight reduction for truck drivers and holds clinical significance in guiding the development of future interventions in this industry. However, additional trials across varied contexts with larger sample populations are needed. 

Main document:

Results: page 10

The majority of the effects observed are medium to large size, indicating likely clinical significance.

Added into discussion page 23:

The clinical significance of these results is important to note, as even a 1-kilogram reduction in weight can reduce the risk of diabetes by 16% [52]. With any reduction in risk of long-term health conditions developing, we are supporting the health and wellbeing of truck drivers. Only 2 studies reported a significant difference with a small effect (so there is the possibility that the effect is not clinically significant), whereas 5 of the differences identified were a medium to large effect, suggesting the differences observed are likely to be clinically significant and meaningful. Using a multi-pronged method of delivery as described in the studies with the most efficacious findings in this review, is more likely to yield effective long-term results, however, many of the studies were small in number with varied effect sizes and additional research is still required.

- Is there a typo in the following sentence? I am not sure what it is trying to say: “The effect of assignment regarding was the following outcomes (weight, BMI, fat mass, body measurements), converting all to kilograms (kg) and centimeters (cm) for ease of comparison, reporting effect sizes between baseline and reassessment 140 and/or follow-up if provided.” 

 Changed – page 7 to “All outcome measurements (weight, BMI, fat mass, body measurements), were converted to kilograms (kg) and centimeters (cm) for ease of comparison, reporting effect sizes between baseline and reassessment and/or follow-up if provided.”

- Discussion: “This systematic review has identified seven intervention studies conducted over the past 20 years that explored the effect of weight loss interventions on obese truck drivers.” I don’t think the latter part of the sentence is completely accurate since the inclusion criteria did not specify that the truck drivers had to have obesity, rather the review focuses on studies of lifestyle interventions for truck drivers in general, regardless of their starting weight. In other words, I suggest removing the word ‘obese’ from this sentence. Removed the word obese – page 19

Reviewer #2: The author(s) have sufficiently addressed each of my comments. I have no further comments other than to thank the authors for the opportunity to read and review their work.

Thank you again for this opportunity, and we hope we have addressed all the remaining concerns and comments.

Kind regards

Dr Elizabeth Pritchard

On behalf of the research team

---

## [Editor Report · Decision Letter 2]

10 Jan 2022

The effect of weight loss interventions in truck drivers: Systematic review

PONE-D-21-02019R2

Dear Dr. Pritchard,

We’re pleased to inform you that your manuscript has been judged scientifically suitable for publication and will be formally accepted for publication once it meets all outstanding technical requirements.

Kind regards,

Lisa Susan Wieland

Academic Editor

PLOS ONE

---

## [Editor Report · Acceptance letter]

31 Jan 2022

PONE-D-21-02019R2 

The effect of weight loss interventions in truck drivers: Systematic review

Dear Dr. Pritchard:

I'm pleased to inform you that your manuscript has been deemed suitable for publication in PLOS ONE. Congratulations! Your manuscript is now with our production department. 

Kind regards, 

on behalf of

Dr. Lisa Susan Wieland 

Academic Editor

PLOS ONE